# An Interpretable Time Series Data Prediction Framework for Severe Accidents in Nuclear Power Plants

**DOI:** 10.3390/e25081160

**Published:** 2023-08-02

**Authors:** Yongjie Fu, Dazhi Zhang, Yunlong Xiao, Zhihui Wang, Huabing Zhou

**Affiliations:** 1College of Computer Science and Engineering, Wuhan Institute of Technology, Wuhan 430205, China; fuyongjie1217@163.com; 2Hubei Provincial Key Laboratory of Intelligent Robot, Wuhan Institute of Technology, Wuhan 430205, China; 3CNNC Key Laboratory on Nuclear Industry Simulation, China Nuclear Power Operation Technology Corporation, Ltd., Wuhan 430040, China; zhangdz02@cnnp.com.cn (D.Z.); ablezhh9703@163.com (Z.W.); 4China Nuclear Power Operation Technology Corporation, Ltd., Wuhan 430040, China; xiaoyl01@cnnp.com.cn

**Keywords:** time series prediction, GRU, SHAP, MSLB, LOCA

## Abstract

Accurately predicting severe accident data in nuclear power plants is of utmost importance for ensuring their safety and reliability. However, existing methods often lack interpretability, thereby limiting their utility in decision making. In this paper, we present an interpretable framework, called GRUS, for forecasting severe accident data in nuclear power plants. Our approach combines the GRU model with SHAP analysis, enabling accurate predictions and offering valuable insights into the underlying mechanisms. To begin, we preprocess the data and extract temporal features. Subsequently, we employ the GRU model to generate preliminary predictions. To enhance the interpretability of our framework, we leverage SHAP analysis to assess the contributions of different features and develop a deeper understanding of their impact on the predictions. Finally, we retrain the GRU model using the selected dataset. Through extensive experimentation utilizing breach data from MSLB accidents and LOCAs, we demonstrate the superior performance of our GRUS framework compared to the mainstream GRU, LSTM, and ARIMAX models. Our framework effectively forecasts trends in core parameters during severe accidents, thereby bolstering decision-making capabilities and enabling more effective emergency response strategies in nuclear power plants.

## 1. Introduction

With the rapid growth in the social economy, traditional thermal power generation fails to meet societal needs and results in irreversible environmental pollution. The large demand for energy in various countries makes the vigorous development of nuclear energy inevitable [1]. Additionally, promoting the development of nuclear energy is one of the essential measures for China to adjust its energy structure and achieve the goal of being “dual carbon” [2]. However, in the nuclear energy field, nuclear power plants’ operational safety is highly critical, and any mistake may lead to catastrophic consequences. Therefore, to ensure the safe and reliable operation of nuclear power plants, the diagnosis and prediction of nuclear power operating conditions must be accurate and efficient, and it is essential to study the prediction of severe nuclear power accidents [3].

A loss of coolant accident (LOCA), main steam line break (MSLB) accident, and steam generator tube rupture (SGTR) accident are critical accidents that pose safety threats to nuclear power plants. Predicting the progression of these accidents is a crucial emergency measure to assess safety risks in advance and enable timely and effective countermeasures. However, their prediction is challenging due to the transient nature and complex characteristics of nuclear power accidents. The data involved are a nonstationary time series with nonlinear and noisy characteristics, further complicated by the influence of temporal and characteristic dimensions. Thus, developing a forecasting model that considers complex systems, nonlinear parameters, and multiple operating nodes remains a key research focus in time series data prediction [4].

To enhance forecast accuracy, researchers initially employed traditional models such as autoregressive integral moving average (ARIMA) [5] and support vector regression (SVR) [6,7] to explore nuclear power forecasting. In the following years, artificial neural networks (ANNs) [8], convolutional neural networks (CNNs) [9], recurrent neural networks (RNNs), long short-term memory networks (LSTMs), gated recurrent units (GRUs) [10], etc., along with their variants, were employed to improve prediction accuracy.

Improvements in model prediction accuracy often come at the expense of increased model complexity. However, increased complexity can result in reduced explanatory power of the prediction outcomes, compromising the credibility and practicality of the model. The highly nonlinear and abstract characteristics of deep learning models pose challenges for intuitively comprehending the decision-making process. The models are regarded as “black box” models due to the opaque nature of the models’ internals. Consequently, existing deep learning methods suffer from a lack of interpretability. This falls short of meeting the rigorous safety standards required in nuclear power plants. Therefore, in terms of practical engineering deployment, pursuing model interpretability represents a critical challenge for artificial intelligence models at this stage [11].

To tackle these challenges, we integrated the GRU and SHAP methodologies to construct a framework called GRUS. This framework aims to forecast and elucidate the trends of severe accidents in nuclear power plants, and it establishes a trusted relationship between users and models, effectively mitigating potential threats of models in the deployment and application of nuclear power. The key contributions of this study can be summarized as follows.

(1) We propose a hybrid framework called GRUS for predicting the trends of severe accidents in nuclear power plants, achieving a higher prediction accuracy. Furthermore, the interpretability analysis of the prediction model provides a more reliable and comprehensive reference and practical value.

(2) We pioneer the utilization of the SHAP algorithm for the interpretability analysis of nuclear power severe accident time series prediction. This approach leads to a deeper understanding of key features that have a significant impact on predictive models, thereby improving the effectiveness and efficiency of forecasting.

(3) We conduct a comparative analysis, evaluating the performance of our proposed GRUS model against well-established GRU, LSTM, and classic ARIMAX models. The results conclusively demonstrate the superiority of our model over these state-of-the-art comparison models.

## 2. Related Work

A comprehensive review of relevant research indicates that the development of artificial intelligence (AI) has facilitated its utilization in the simulation and analysis of operational data, and has received increasing attention from the nuclear industry over the past few decades, resulting in successful applications in nuclear power plants. Despite the increasing application of AI models, interpretability remains a nascent research field, with further development required. Specifically, interpretability analysis is of paramount importance to ensure the safe and effective deployment of these models. Therefore, there is a pressing need for future research in nuclear power plants to focus on enhancing the accuracy of models and integrating interpretability analysis.

### 2.1. Artificial Intelligence Method

Researchers in the past have utilized conventional statistical modeling methods to forecast the functioning of nuclear power plants. For instance, Wu et al. [5] applied the ARIMA model to predict the levels of radioactive substances in the vicinity of nuclear power plants. Moshkbar et al. [12] developed a robust accident recognition system that integrates the ARIMA model and the EBP algorithm for identifying and monitoring the transient processes during nuclear power plant accidents. With the advancements in artificial intelligence, researchers have started employing neural networks to model and predict the operational data of nuclear power plants. Santhosh et al. [13] proposed a data-driven nuclear reactor accident state identification method based on the reactor process parameters using an ANN, which can effectively diagnose potentially threatening accident situations. Kim et al. [14] proposed a method based on a fuzzy neural network (FNN) to predict the leakage flow rate of an LOCA. Do et al. [15] used a deep neural network (DNN) to achieve the water level prediction of a nuclear reactor pressure vessel. Tian et al. [16] proposed an approach for detecting an LOCA in nuclear power plant condition monitoring using a neural network architecture optimization method based on the constrained random search algorithm. Sun et al. [17] employed principal component analysis (PCA) to identify an SGTR occurring in the steam generator tubes of a small modular reactor and to quantify the size of the fracture. In addition, Wang et al. [18] introduced the fault identification and diagnosis method based on kernel principal component analysis (KPCA) and similarity clustering for sensor anomaly diagnosis in nuclear power plants. Yang et al. [19] used the RELAP5/MOD3.3 code to simulate the MSLB of Gen III reactors with passive safety functions, and proposed an optimal estimation plus uncertainty analysis method, which can analyze and evaluate the MSLB event in Gen III reactors. Xiang et al. [20] introduced an unsupervised representation clustering method based on deep learning for the automatic transient pattern recognition of nuclear power plant condition monitoring data.

A CNN is a variation of a DNN that is commonly employed in various fields such as natural language processing and image processing. Because of its ability to extract valuable features from data properties, a CNN exhibits high robustness for large-scale and complicated datasets. Yao et al. [21] applied a small-batch-size processing CNN to fault diagnosis problems in nuclear energy production, including detecting fault types such as cracks, sensor failures, pipeline corrosion, etc. Lin et al. [22] converted the input sensor measurement time series data into a picture form, and classified and reconstructed the signal based on a CNN to deal with the problem of sensor failure in nuclear power plants. He et al. [23] proposed a data-driven adaptive fault diagnosis method based on non-dominated sorting genetic algorithm II (NSGAII) and a CNN to improve the accuracy and efficiency of fault diagnosis in nuclear power systems.

In terms of the real-time detection and trend prediction of nuclear power plant operating conditions, an RNN is good at processing time series data, and has been successfully employed to sequence data modeling in previous explorations. Şeker et al. [24] presented the study of an RNN for the condition monitoring of nuclear power plants and rotating machinery. LSTM as a variant of traditional RNNs, has attracted much attention due to its high-precision prediction of time series data. Radaideh et al. [25] used an expert system based on a DNN and LSTM to accurately predict an LOCA in nuclear power plants. Nguyen et al. [26] proposed a multi-step time series signal prediction method based on ensemble empirical mode decomposition (EEMD) and LSTM, which was applied to data analysis in nuclear power plants. She et al. [27] constructed a combination of a CNN and LSTM named ConvLSTM for fault diagnosis and the post-accident prediction of an LOCA. Zhang et al. [28] proposed a fault diagnosis model based on a CNN-LSTM neural network optimized by the sparrow search algorithm, which achieved accurate and real-time fault diagnosis and prediction in nuclear power plants. Gong et al. [29] introduced a zigmoid-based LSTM method for the time series prediction of an LOCA. Lei et al. [30] developed a prediction model for key parameters affected by sensor failures during nuclear power plant accidents by using LSTM and historical key parameter operation sequences. The proposed model successfully predicted parameters including the pressurizer pressure, pressurizer water level, and steam generator water level, etc. in the case of an LOCA and SGTR. In comparison with the LSTM model, the GRU model simplifies the design of the gating mechanism, has a faster training speed and fewer parameters, and can effectively process long sequence data. As a result, it exhibits a superior performance in applications. Zhang et al. [31] proposed a GRU-based method for reconstructing the pressurizer water level, which can monitor and predict the pressurizer water level of marine pressurized water reactors. Wang et al. [32] proposed an advanced fault diagnosis method for nuclear power plants based on a convolutional gated recurrent unit (CGRU) and enhanced particle swarm optimization (EPSO). Fukun et al. [33] introduced an LOCA accident state prediction method for nuclear power plants based on a GRU-CNN, which can effectively predict the LOCA accident status. He et al. [34] used LSTM and GRU algorithms to construct a U-tube steam generator water level prediction model, which can more accurately predict the change trend of the water level.

Through a comprehensive examination of the recent literature in the field of nuclear power plants, this study chose the current mainstream GRU model as the primary method for accurate nuclear power accident prediction.

### 2.2. Interpretability Method

AI methods have demonstrated remarkable success in analyzing predictive models of structured datasets. However, transparency and interpretability are essential requirements for model development. In nuclear power scenarios and other industrial situations, most models are regarded as “black box” models due to their limited transparency. For instance, the construction of a model for predicting the development trend of a variable in an LOCA relies on historically related parameter values to obtain a future predicted value for the variable. However, the model’s internal prediction mechanism cannot be explained, ultimately reducing the credibility and practicability of the model. To address these limitations and ensure safe engineering deployment when applying AI models in nuclear power plants, explainable artificial intelligence (XAI) has gained increasing attention in recent research endeavors. In this context, XAI is concerned with the development of methods that aim to enhance the models’ interpretability and facilitate operators’ comprehension. Kim et al. [35] systematically summarized the application and problems associated with XAI technology in the domain of operator assistance systems, and provided valuable references to guide future research in the field of nuclear energy. Mortenson et al. [36] conducted a comprehensive review of the application of AI in the nuclear power field, focusing on the analysis of research related to the concept of interpretability in AI systems. The interpretability and transparency of models are particularly important in critical infrastructure domains such as nuclear power, where both staff and the general public demand high levels of reliability from these systems and facilities. Park et al. [37] presented an intelligent diagnosis approach for nuclear power plants based on a GRU-AE autoencoder that extracts and reconstructs the data, followed by processing and analysis of the data using a LightGBM decision tree classifier to achieve the fault classification and diagnosis. Notably, the SHAP algorithm was employed to explain LightGBM model results, thereby enhancing model reliability. Shin et al. [38] used a CNN for the detection and diagnosis of abnormal events in nuclear power plants, and introduced several interpretability analysis methods, including saliency mapping, guided gradient-weighted class activation mapping (Grad-CAM), and SHAP, to enhance the system’s interpretability.

To sum up, practical applications of models in nuclear power and other industrial fields necessitate the incorporation of interpretability. The interpretability of models not only improves the robustness by identifying potential adversarial perturbations that impact prediction results, but also assists decision makers to understand the rationale behind model predictions and verify the potential causal relationships between variables. This study chose the SHAP method as an important method for interpretability analysis.

## 3. Methods

This section commences with an introduction of the GRU model as the fundamental model, while simultaneously introducing the concept of XAI to provide explainable analysis for models. Finally, the GRUS framework proposed in this study and the corresponding evaluation metrics are introduced.

### 3.1. Gated Recurrent Unit (GRU)

GRU model was proposed by Cho et al. [39] for time series that permits the incorporation of different time scales while simplifying calculations. GRU is a type of RNN architecture that can achieve comparable performance to other RNNs, yet is computed more efficiently. GRU has a simpler structure than other RNNs and consists of only two gating mechanisms, namely the reset gate and the update gate, which regulate the retention or deletion of the information at each time step. The update gate manages the amount of data that historical memory information can preserve until the current moment, while the reset gate determines the extent of historical information to forget. Thus the structure of GRU enables it to adaptively capture dependencies from an extensive sequence of data without discarding information from early sections of the sequence. The process for each GRU unit updates its parameters according to the formula below:(1)$$\begin{array}{c} z_{t}=\sigma \left(W_{z}\cdot [h_{t-1},X_{t}]+b_{z}\right), \end{array}$$
(2)$$\begin{array}{c} r_{t}=\sigma \left(W_{r}\cdot [h_{t-1},X_{t}]+b_{r}\right), \end{array}$$
(3)$$\begin{array}{c} {\hat{h}}_{t}=\tanh\left(W_{h}\cdot [r_{t}*h_{t-1},X_{t}]+b_{h}\right), \end{array}$$
(4)$$\begin{array}{c} h_{t}=\left(1-z_{t}\right)*h_{t-1}+z_{t}*{\hat{h}}_{t}, \end{array}$$
where $W_{z}$, $W_{r}$, and $W_{h}$ are weight parameters, $b_{z}$, $b_{r}$, and $b_{h}$ are biases. σ is the sigmoid function, $X_{t}$ is input time step, $h_{t-1}$ is the previous hidden layer state, $h_{t}$ is the output. $r_{t}$ represents the reset gate, $z_{t}$ represents the update gate. As with any other neural network, the gates in GRU are associated with weights and biases whose values are learned during the model training phase. Figure 1 depicts the internal network structure of GRU.

### 3.2. Explainable AI (XAI)

XAI has emerged as a prominent trend in AI. The purpose of an XAI system is to make the behavior of models more intelligible to humans by providing explanations [40]. XAI models are commonly classified into two types: global interpretability and local interpretability [41]. Global interpretability enables analysts to investigate the structure and parameters of a complex model and comprehend how the model works globally. On the other hand, local interpretability examines the model’s individual predictions locally and attempts to comprehend why the model made a particular decision. The interpretability of a predictive model often depends on estimating the contribution of individual characteristics (i.e., independent variables) to the model’s results. This study introduces the SHAP methodology as an interpretability analysis approach for time series prediction models.

SHAP is an explanatory model of additivity constructed by Lundberg [42] that draws inspiration from cooperative game theory. SHAP is used to explain individual and global predictions of the model. It aims to explain the contribution of each input feature to the prediction. Through the computation of Shapley values, which measures the marginal contribution of features towards predictions, SHAP offers crucial insights into the positive and negative impacts of feature values on estimation results at the sample level. The methodology also reveals the interactions between variables and how this relationship is reflected in the model. In addition, SHAP provides visualizations that aid in comprehending the importance of each feature in the model output. As a result, SHAP enhances the interpretability of the model, promoting trust in its predictions. The Shapley value of each feature is calculated as shown in the following equation:(5)$$\begin{array}{c} \phi_{i,j}=\sum_{S\subseteq M\setminus \left\{x_{i,j}\right\}}\frac{|S|!(m-|S|-1)!}{m!}[f_{x_{i}}\left(S\cup \left\{x_{i,j}\right\}\right)-f_{x_{i}}\left(S\right)], \end{array}$$
where *M* is the set of all input features in the dataset with dimension *m*. *S* is a subset of *M* with a size of |S|. $f_{x_{i}}\left(S\right)$ denotes the predicted value of the model for the sample $x_{i}$ when only the feature set *S* is utilized. When *S* is an empty set, $f_{x_{i}}\left(S\right)$ is referred to as the basic value $\phi_{0}$, which equals the average value of the predicted value of the model output among all samples. $f_{x_{i}}\left(S\cup \left\{x_{i,j}\right\}\right)$ is computed by adding the feature value $x_{i,j}$ to the feature set *S*, which yields the average predicted value of the model for the sample $x_{i}$ in all samples. $\frac{|S|!(m-|S|-1)!}{m!}$ represents the coefficient used to normalize the differences. As there are numerous potential values of *S*, it can be inferred from the formula that the Shapley value $\phi_{i,j}$ of the eigenvalue $x_{i,j}$ takes into account all possible combinations of *S*, along with the influences of all other features except $x_{i,j}$. When $\phi_{i,j}$ > 0, it means that the corresponding feature has a positive effect in enhancing the predicted output, otherwise it reduces the contribution to the prediction. SHAP utilizes the feature attribution method to calculate the attribution value of each feature, which reflects its impact on the final prediction. A higher SHAP score indicates a greater contribution to the feature.

### 3.3. GRUS Framework

In this study, we propose an explainable framework GRUS based on the GRU model, which integrates the SHAP methods. GRUS utilizes the relevant features of severe nuclear power accidents and the information to predict the trend of key parameters. Simultaneously, it conducts explainable analysis to facilitate understanding of the model’s prediction mechanism, identification of internal factors influencing nuclear power accidents, and improvement of trust in AI models for nuclear power scenarios among relevant personnel.

The GRUS framework mainly includes two parts: data processing and framework analysis. The data processing component consists of four sequential steps, i.e., feature engineering, data normalization, sliding update, and dataset splitting. The framework encompasses the GRU model for training and prediction, the SHAP method for interpretability analysis, model optimization, and enhancement of prediction results. To elaborate, the following provides a detailed description of the specific implementation process.

#### 3.3.1. Data Processing

Feature Engineering

The aim of feature engineering is to enhance prediction accuracy and accelerate computations by reducing the number of features during model training. Due to the large number of features in the data, a feature correlation heat map is generated to comprehend the features. For feature analysis, Spearman, Pearson, and Kendall correlation coefficients (i.e., SCC, PCC, KCC) are commonly used to evaluate monotonicity, linearity, and dependence of different state parameters, respectively. In this study, we employed Spearman correlation and Pearson correlation to analyze the relationships between features, aiming to select highly correlated features with the target features. By utilizing these two correlation methods, we ensured a comprehensive and precise analysis of the correlations, which facilitated the selection of input variables for our modeling process.

Additionally, the F-score can be employed to select highly relevant features for analysis. By perturbing a column of the feature matrix, the error score of the feature represented by the column to the target feature is regarded as the F-score. The F-score of each feature is calculated as a measure of its discriminatory strength, with higher scores indicating greater discriminative capability and more importance for the feature. All F-scores are computed for each feature to facilitate feature selection. By comparing feature importance, the most effective features can potentially be identified.

Finally, the features are filtered again by incorporating expert experience, resulting in the selection of key system parameters as modeling features.

Data Normalization

Raw data may not be appropriate for model input if their values are too large or small, hindering the model’s ability to learn temporal correlations. Therefore, it is important to normalize input data by scaling it to a specified range. In our experiment, we adopted the MinMaxScaler method to normalize the data, which results in rapid convergence during model training.

Sliding Update

Accurate time series data prediction often necessitates considering the influence of historical data on the succeeding moment. The sliding window technique is a valuable method for processing data to maximize the utilization of all data.

The sliding window technique retains a fixed time step and uses the initial *k* data points as inputs for the model. The k+1th data point serves as the label, representing the actual value of the time series prediction for that data group. As new data arrive, the sampling window slides and updates, appending the latest data while removing historical data. This method is widely used for processing time series data because it efficiently captures temporal dependencies and improves forecasting accuracy. Its schematic diagram is shown in Figure 2.

Splitting Dataset

The datasets are divided into sub-datasets of the training set, validation set, and test set, which are utilized for model training, model verification, and model testing, respectively. Based on the sub-datasets, we can obtain a well-trained GRUS to predict the time series trend of some key parameters in nuclear power accidents.

#### 3.3.2. Framework Analysis

After data preprocessing, the GRUS framework is employed to predict time series data and conduct interpretability analysis. The GRU model is leveraged to predict time series data by capturing the nonlinearity of fluctuating features and controlling information flow through the network via gating mechanisms to achieve accurate forecasting. Subsequently, SHAP analysis is conducted to analyze the model’s interpretability and comprehend its prediction mechanism, enabling optimization of the model at the feature level by selecting important relevant features in the interpretability analysis, so as to achieve the improvement in the prediction performance. Ultimately, prediction results are generated and corresponding evaluation metrics are utilized for evaluation and comparison. The flowchart of the GRUS framework is shown in Figure 3.

### 3.4. Metrics

In order to evaluate the performance of the prediction model, this study employed three commonly utilized evaluation metrics in regression modeling, namely, mean square error (MSE), mean absolute error (MAE), and mean absolute percentage error (MAPE). Smaller MSE and MAE values indicate higher prediction precision, the range of MAPE is [0, +∞), 0% means a perfect model, and 100% means an inferior model. The formulas for these metrics are as follows:(6)$$\begin{array}{cc} MSE & =\frac{1}{N}\sum_{i=1}^{N}{\left({\hat{y}}_{i}-y_{i}\right)}^{2}, \end{array}$$
(7)$$\begin{array}{cc} MAE & =\frac{1}{N}\sum_{i=1}^{N}\left|\left({\hat{y}}_{i}-y_{i}\right)\right|, \end{array}$$
(8)$$\begin{array}{c} MAPE=\frac{\sum_{i=1}^{N}\left|{\hat{y}}_{i}-y_{i}\right|/y_{i}}{N}\times 100\%, \end{array}$$
where *N* represents the number of samples in the dataset, $y_{i}$ represents the actual value, ${\hat{y}}_{i}$ represents the predicted value generated by the evaluated model.

## 4. Experiment and Results

This section starts with a detailed description of the experimental datasets utilized in the study. Then, the data are subjected to rigorous processing for subsequent analyses. Additionally, this section introduces the relevant experimental settings, which include the both qualitative and quantitative experiments to evaluate the performance of the proposed GRUS framework. Finally, the experimental outcomes are meticulously presented and elaborately analyzed in detail to facilitate a comprehensive understanding of the research findings.

### 4.1. Data Source

The experiment used the Modular Accident Analysis Program version 5 (MAAP5), an industrial-grade nuclear power simulation platform, to simulate severe nuclear power accidents. As an integrated system model, MAAP5 can simulate the response of the reactor core, reactor coolant system, and containment under conditions of core damage. The experiment simulated two typical accident scenarios of an LOCA and MSLB at 100% reactor power, with the accident pipe rupture size set to 0.5 m^2^. The resulting datasets contained 116 features divided into 10 categories, including the main coolant system, passive safety system, steam generator parameters, confinement parameters, primary coolant system parameters, and essential main system signals, among others.

### 4.2. Data Processing and Analysis

In order to further screen important feature parameters, we calculated and drew the PCC and SCC heatmap between all features in both types of accidents using feature engineering techniques. Then we were able to select the relevant features with the total power of the reactor (QCORET) based on expert experience. Figure 4 and Figure 5 show the heat maps for the Pearson and Spearman correlation analyses of some features, respectively.

Additionally, feature selection was conducted using the F-Score to identify high-relevance features specific to QCORET for further screening of the features. Figure 6 depicts the F-Scores of some features.

Finally, feature screening based on expert knowledge was employed to further refine the selection, resulting in the identification of 25 key system parameters that were to be utilized as modeling features. The screening features are presented in Table 1.

### 4.3. Experimental Setting

Following the application of data processing techniques, including filtering the original dataset and normalizing the data with a MinMaxScaler method, a sliding window mechanism with a step size of 50 was set to roll and split the data. By using these processed datasets, models were developed to predict the progression of two different types of severe nuclear power accidents, namely an MSLB and LOCA.

This study proposes a GRUS model that uses the historical data of an MSLB and LOCA to predict the respective development trends of the two nuclear power accidents. The model is optimized by tuning hyperparameters, including the number of network layers, the number of neurons in each hidden layer, the learning rate of the optimizer, and the batch size used during training. Through the training and optimization of the GRUS model, the optimal parameters are obtained. The GRUS model in this study was constructed with a structure of 1 input layer, 3 hidden layers, and 1 output layer, where each hidden layer had 256 neurons. The optimal parameters are shown in Table 2.

There are two types of experiments, quantitative evaluation and qualitative evaluation. The quantitative evaluation experiment involves the comparison of the GRUS model framework with both ARIMAX, LSTM, and GRU, aimed at verifying its accuracy effect. In the qualitative evaluation experiment, given the lack of interpretability of the model, an interpretability analysis of SHAP in the GRUS framework is performed to shed light on the mechanism underlying the model’s predictive performance. This approach facilitates a more thorough exploration of the model and improves its overall interpretability.

### 4.4. Experimental Results

#### 4.4.1. Quantitative Evaluation

The experiment conducted a quantitative analysis to verify the accuracy of the GRUS framework on the time series prediction of nuclear power accident scenarios. The prediction performance of the GRUS was compared with that of the ARIMAX, LSTM, and GRU models using evaluation metrics such as MSE, MAE, and MAPE through quantitative analysis. The primary loop flow (i.e., WLOOP(1)), which is the most affected parameter during MSLB accidents, was chosen as the first key predictive feature, whereas pressurizer pressure (i.e., PPZ) was selected as the predictive feature for LOCAs. The experimental results are shown in Table 3, with the data in bold representing the optimal values after comparison.

Table 3 presents the evaluation results of the GRUS framework, and the ARIMAX, LSTM, and GRU models for predicting the time series data related to LOCAs and MSLB accidents. The GRUS framework outperforms the other models in terms of all evaluation metrics, including MSE, MAE, and MAPE.

Figure 7 and Figure 8 show the predicted results of the GRUS and other models taking a continuous 100 s of data after the accident. As can be observed from these figures, the predicted values of GRUS are much closer to the actual values, as indicated by the lower absolute differences between the predicted and actual values. Therefore, the GRUS framework can provide more accurate predictions and is a suitable choice for predicting severe nuclear power accidents due to its superior predictive capabilities.

#### 4.4.2. Qualitative Evaluation

The experiment conducted a qualitative analysis on the SHAP part of the GRUS framework. From the two perspectives of local and global interpretability, the SHAP index distribution of different features on a single sample and global samples was visualized to reveal the importance of the support of various features in the dataset to the prediction results. Additionally, an interpretability analysis was performed on the feature dependencies extracted from the prediction model to improve the prediction performance of the model.

Global Interpretability

Taking the MSLB accident as an example, Figure 9 shows an analysis of feature importance using SHAP in the GRU model of the GRUS framework, which can understand the importance of features in the neural network model. The first five important influencing factors were investigated, with WLOOP(1) identified as having the most significant impact on the model because the primary loop flow changes rapidly following the rupture of the main steam pipe. Other influential features include TCL1, TGCH(1), TGRB(4), and WLOOP(3), suggesting that parameters such as temperature also undergo significant changes. It can be seen that during the training process of the model, the neural network nodes focus on relevant feature nodes to realize accurate predictions of future results.

Figure 10 shows the distribution of the SHAP value of each influencing factor. Each data point on each feature stream in the figure represents each instance. The SHAP value of each feature determines the position of the point on the X-axis, the points along the feature rows are stacked to show the density, and the color indicates the value of the feature in the original dataset, from blue to red to indicate the change in the value from low to high. A positive SHAP value indicates that the contribution to the target variable is positive, and a negative SHAP value suggests that the contribution is negative. In addition, the variable’s value decreases as the points on the graph move closer to being blue and increases as they move closer to being pink. For instance, if the air temperature in the upper compartment of the containment is raised (i.e., TGRB(4) is increased), the corresponding SHAP value of TGRB(4) also increases. This indicates that an increase in the air temperature of the upper compartment of the containment vessel caused by a rupture in the main steam pipe, produces a subsequent increase in the temperature of the coolant on the primary circuit side of the steam generator, thereby elevating the water vapor pressure inside the primary circuit. In order to safeguard the safety and stability of the reactor core, the flow rates and pressures within the system are precisely regulated via the control system. Consequently, the primary loop flow usually increases with the increase in the temperature in the upper compartment of the containment vessel, showing that the characteristic TGRB(4) is positively correlated with the predicted target WLOOP(1). However, the SHAP analysis suggests that WLOOP(1) is negatively correlated with the temperature of the cold pipe section (i.e., TCL1). Specifically, as the temperature of TCL1 decreases, its SHAP value will decrease, resulting in an increase in the predicted target. This can be explained by the fact that a decrease in the temperature of the cold pipe section results in an increase in the density of the coolant, leading to a reduction in the flow resistance and ultimately causing an elevation in the flow rate within the pipe. To sum up, the ability of the model to accurately predict outcomes can be attributed to its capacity for learning the interdependencies between features during the training process, allowing it to acquire a comprehensive understanding of feature interactions.

Local Interpretability

Figure 11 shows a SHAP dependency plot, which illustrates the impact of the observation’s variable values on the predicted result. The dependency plot enables a comprehensive description of both the major effects of individual observations and their interactions. The left figure in Figure 11 demonstrates that an increase in the temperature of the cold pipe section leads to a corresponding decrease in its SHAP value and contribution, resulting in a decrease in the estimated target WLOOP(1). It shows an inverse relationship between WLOOP(1) and TCL1, with a corresponding negative correlation between the two variables. The right figure depicts the connection between the air temperature in the top compartment of the containment and the primary loop flow rate. As TGRB(4) gradually increases, the positive correlation between TGRB(4) and WLOOP(1) becomes stronger.

The model provides predictions based on the most significant chosen attributes, as demonstrated by the utilization of the SHAP approach for interpretability analysis. The identified relationship between the chosen input features and the predicted output variable enables robust conclusions to be drawn from the model’s analysis and application. Furthermore, the model can be further optimized by removing less important features, improving the speed and efficiency of the prediction process.

## 5. Conclusions

In the event of a severe nuclear power accident, accurately predicting real-time operating conditions can transform nuclear power safety control from reactive measures to proactive intervention. Furthermore, incorporating interpretability analysis to comprehend the results of AI models enhances the trust of the operation and maintenance personnel, which is essential in providing effective decision support for nuclear power emergency response strategies.

This study proposes an interpretable hybrid framework that combines GRU and SHAP to address the challenges of accurately predicting key parameters in nuclear power accidents. By analyzing two types of accidents, an MSLB and an LOCA, we demonstrate the successful application of this model. The experiment carries out quantitative analysis and qualitative analysis. By comparing it with the ARIMAX, LSTM, and GRU models, it is shown that the GRUS framework is superior to other baseline models in nuclear power accident prediction. The evaluation of the two accident indexes results in MSEs of 8.6385$\;\times \;10^{-6}$ and 2.1881$\;\times \;10^{-5}$, MAEs of 1.7704$\;\times \;10^{-3}$ and 3.2231$\;\times \;10^{-3}$, and MAPEs of 0.0373% and 0.0587%. Simultaneously, the SHAP framework provides an explanation of the model and analyzes the influence of variables on different accidents, identifying WLOOP(1), TCL1, TGCH(1), and TGRB(4), etc. as indicators with significant impacts in the MSLB accident. Moreover, the correlation between TCL1 and TGRB(4) and the predicted target primary loop flow WLOOP(1) is analyzed. These results can enhance nuclear power workers’ understanding of the relationship between complex parameter indicators and facilitate decision making by optimizing resource utilization through the consideration of fewer features. Furthermore, these results enable the safety system to intervene proactively before the actual accident occurs, thereby mitigating the potential for more severe losses.

Several issues require further research beyond this work. In severe nuclear power accidents, it is necessary to utilize diverse breach accident data for analysis; as our experiment solely considers fixed breach data in two types of accidents, additional accident data can supplement future analyses. Additionally, real accident data from nuclear power plants can be utilized as a complement to this study if available. Furthermore, in terms of interpretability analysis, other explanation methods can be employed to model predictions to enhance the effectiveness of the model. Therefore, future research should explore and address these issues to augment nuclear power safety control measures.

## Figures and Tables

**Figure 1 entropy-25-01160-f001:**
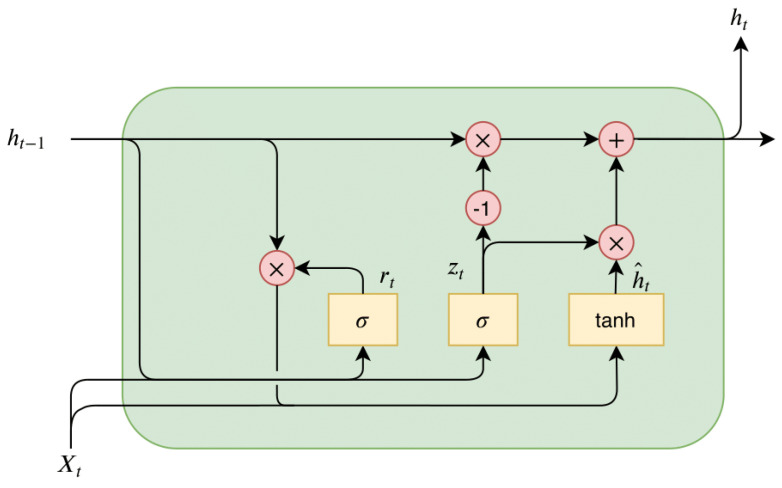
The internal structure of one neural unit of GRU.

**Figure 2 entropy-25-01160-f002:**
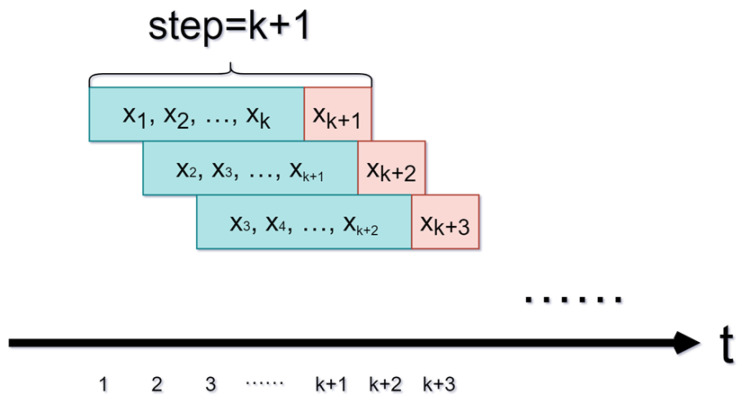
The sliding window technique.

**Figure 3 entropy-25-01160-f003:**
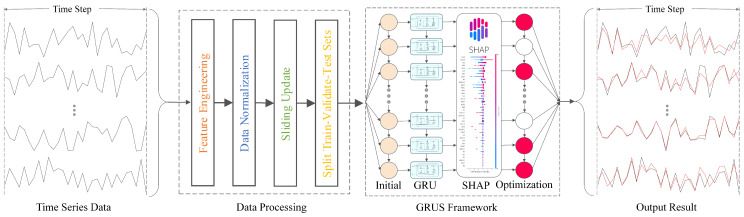
The flow–chart of the GRUS framework.

**Figure 4 entropy-25-01160-f004:**
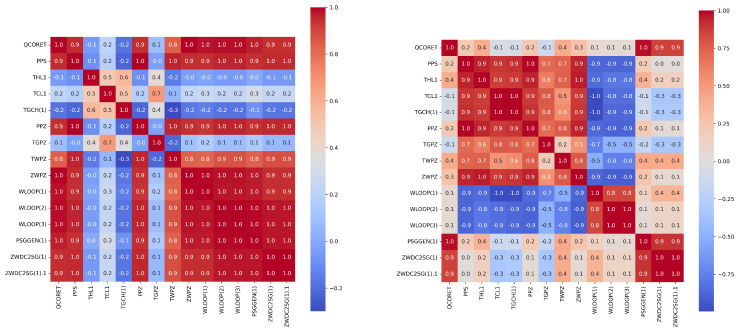
The Pearson correlation coefficients heat–map of some features of the LOCA (**left panel**) and the heat–map of some features of the MSLB accident (**right panel**).

**Figure 5 entropy-25-01160-f005:**
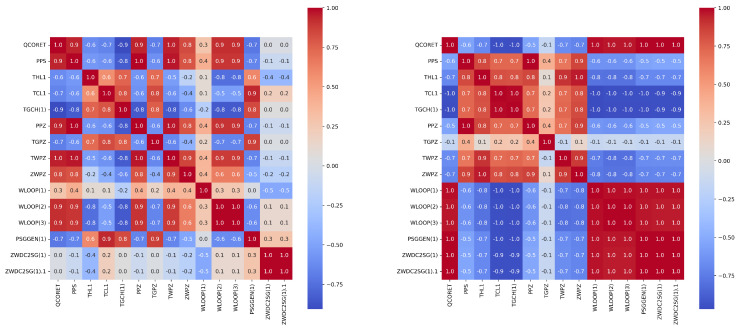
The Spearman correlation coefficients heat–map of some features of the LOCA (**left panel**) and the heat–map of some features of the MSLB accident (**right panel**).

**Figure 6 entropy-25-01160-f006:**
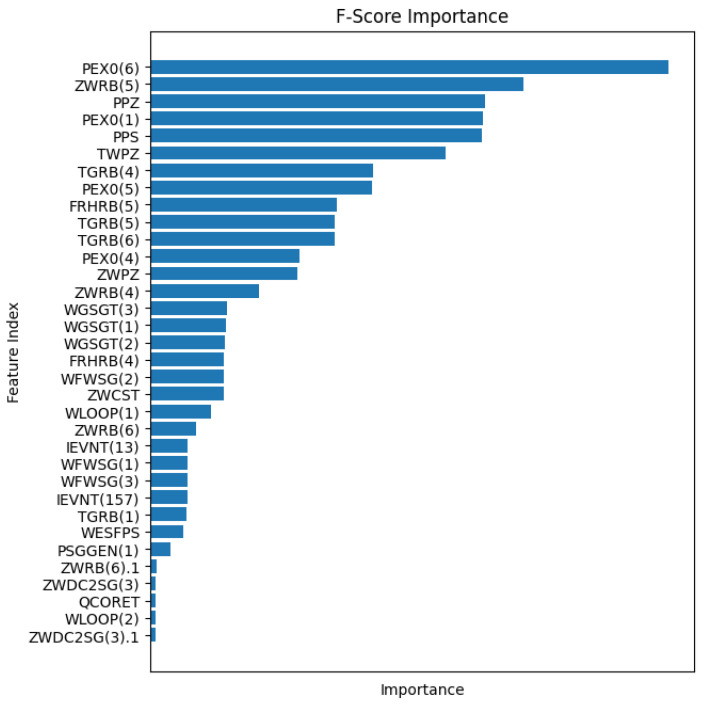
The F-Score of some features.

**Figure 7 entropy-25-01160-f007:**
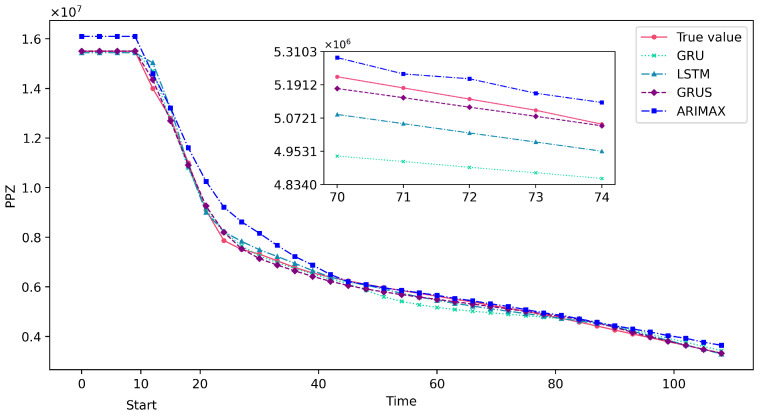
Partial prediction results of GRUS and other models on the test data in the LOCA.

**Figure 8 entropy-25-01160-f008:**
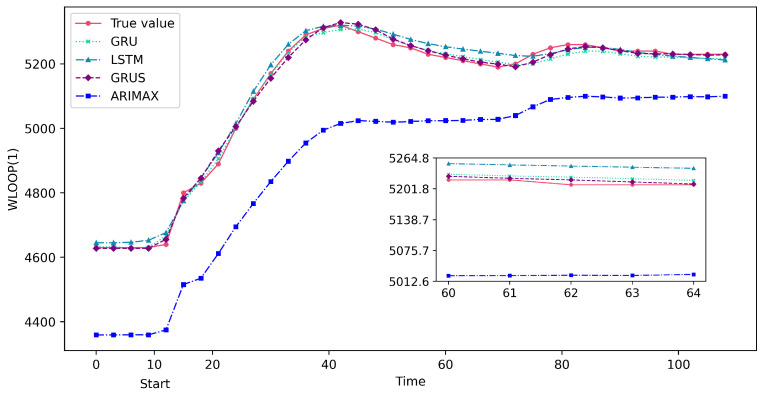
Partial prediction results of GRUS and other models on the test data in the MSLB accident.

**Figure 9 entropy-25-01160-f009:**
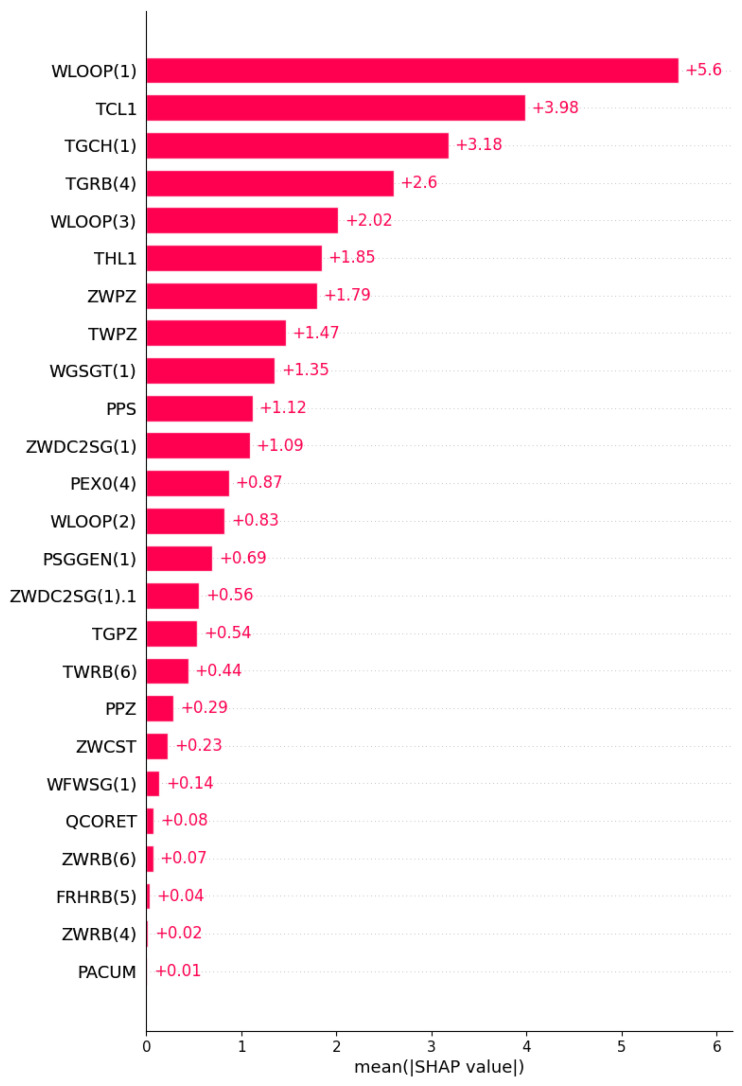
The importance of input features in the SHAP.

**Figure 10 entropy-25-01160-f010:**
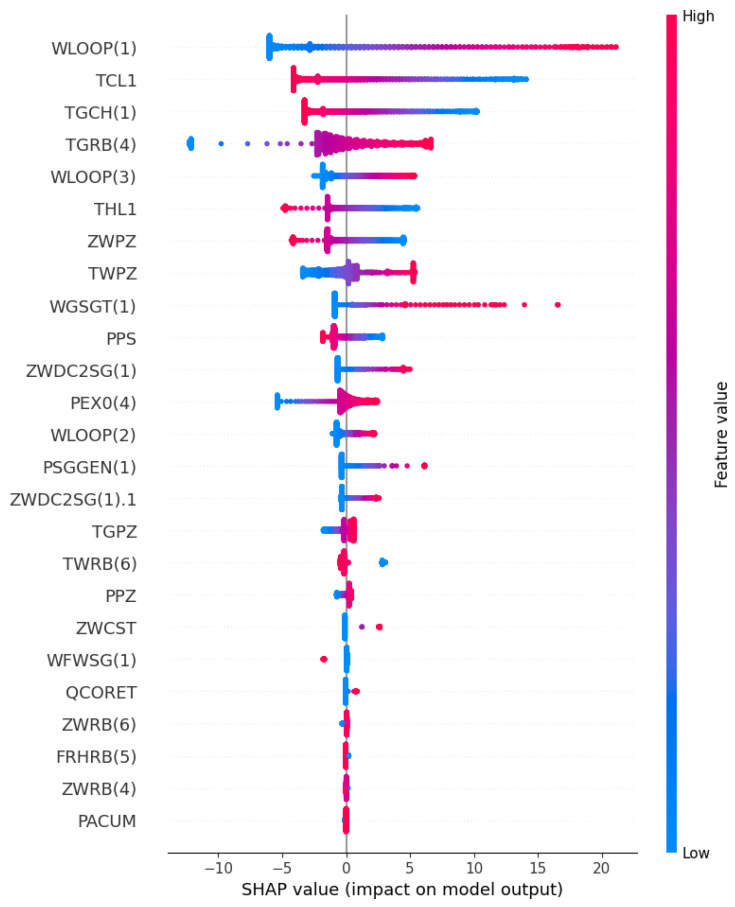
The beeswarm plot of input features in the SHAP.

**Figure 11 entropy-25-01160-f011:**
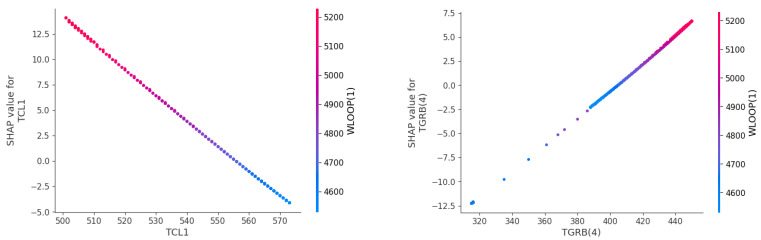
The dependency plot of TCL1 analysis in the SHAP (**left panel**). The dependency plot of TGRB(4) analysis in the SHAP (**right panel**).

**Table 1 entropy-25-01160-t001:** The screening features by feature engineering.

Feature Name	Feature Name	Feature Name	Feature Name
QCORET	TWPZ	ZWDC2SG(1)−W	ZWRB(4)
PPS	ZWPZ	ZWDC2SG(1)−N	FRHRB(5)
THL1	WLOOP(1)	WFWSG(1)	ZWRB(6)
TCL1	WLOOP(2)	WGSGT(1)	TWRB(6)
TGCH(1)	WLOOP(3)	PEX0(4)	ZWCST
PPZ	PSGGEN(1)	TGRB(4)	PACUM
TGPZ			

**Table 2 entropy-25-01160-t002:** Optimal parameter settings for the GRUS framework.

Parameters	Values
Learning Rate	0.0001
Number of input layer units	25
Number of hidden layers	3
Number of neurons	256
Batch_size	32
Number of epochs	300
Optimizer	Adam
Dropout	0.2

**Table 3 entropy-25-01160-t003:** Performance comparison of the models’ prediction results.

Scenarios	Model	MSE	MAE	MAPE
MSLB	ARIMAX	$4.6798\;\times \;10^{-3}$	$5.7732\;\times \;10^{-2}$	2.2553%
	LSTM	$1.7419\;\times \;10^{-4}$	$1.2245\;\times \;10^{-2}$	0.3077%
	GRU	$4.1627\;\times \;10^{-5}$	$5.0610\;\times \;10^{-3}$	0.0862%
	GRUS	$2.1881\;\times \;10^{-5}$	$3.2231\;\times \;10^{-3}$	0.0587%
LOCA	ARIMAX	$1.0805\;\times \;10^{-4}$	$6.6404\;\times \;10^{-3}$	0.2004%
	LSTM	$6.8415\;\times \;10^{-5}$	$6.8512\;\times \;10^{-3}$	0.5112%
	GRU	$3.7715\;\times \;10^{-5}$	$4.7580\;\times \;10^{-3}$	0.3553%
	GRUS	$8.6385\;\times \;10^{-6}$	$1.7704\;\times \;10^{-3}$	0.0373%

## Data Availability

The data are available from the corresponding author of this paper.

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
