# Peer review of "An Interpretable Time Series Data Prediction Framework for Severe Accidents in Nuclear Power Plants"

_entropy, 2023, doi:10.3390/e25081160_

Round 1

Reviewer 1 Report

Major remarks:

- Why did you use both Spearman and Pearson correlations? Could you explain how did you use them? The using of correlation measures depends on feature character. Additionally, we are looking for features highly correlated with the response and weakly correlated with each other. What is the response in the experiments?

- Could you explain in detail how did you calculate F-score?

- Using R^2 as a measure of quality for the non-linear model is questionable. I propose to replace it with MAPE which is complementary to MAE.

- How did you select parameters for the model (Table 2)?

- In my opinion, it is worth adding the classic model ARIMAX to the comparison. 

- Differences in results are rather small. Do you think that they are important from a practical point of view?

Minor remarks:

- At the end of most formulas should be a comma or full stop.

- We enumerate only equations that we cite in the text.

Reviewer 2 Report

This paper is not outstanding but reasonable in terms of significance, novelty, meaningfulness of experimental results. 

The methodology is straightforward and result presentation are in the norm of similar papers. There is comparative study with some SOTA work.

As authors point out it is a critical area, the prediction accuracy is expected to be robust. The existing deep learning methods suffer from lack of explanability, which is mentioned but not well articulated. 

Ok

Round 2

Reviewer 1 Report

Authors made a huge effort to correct and improve their paper. They responded to all raised problems and I appreciate this. Finally, I can recommend this paper for publication.